# Clinical manifestations and health outcomes associated with Zika virus infections in adults: A systematic review

Sheliza Halani[1], Panashe E. Tombindo[1], Ryan O'Reilly[1,2], Rafael N. Miranda[2], Laura K. Erdman[1,3], Clare Whitehead[1,4,5,6], Joanna M. Bielecki[2], Lauren Ramsay[1,2], Raphael Ximenes[2,7], Justin Boyle[1], Carsten Krueger[1,3], Shannon Willmott[1,3], Shaun K. Morris[1,3,8], Kellie E. Murphy[1,4], Beate Sander[1,2,9,10]*

1 University of Toronto, Toronto, Ontario, Canada, 2 Toronto Health Economics and Technology Assessment (THETA) Collaborative, University Health Network, Toronto, Ontario, Canada, 3 Division of Infectious Diseases, Hospital for Sick Children, Toronto, Ontario, Canada, 4 Department of Obstetrics and Gynaecology, Mount Sinai Hospital, Toronto, Ontario, Canada, 5 Department of Obstetrics and Gynaecology, University of Melbourne, Parkville, Australia, 6 Pregnancy Research Centre, The Royal Women's Hospital, Parkville, Victoria, Australia, 7 Escola de Matemática Aplicada, Fundação Getúlio Vargas, Praia de Botafogo, Rio de Janeiro, Brasil, 8 Centre for Global Child Health, Hospital for Sick Children, Toronto, Ontario, Canada, 9 Public Health Ontario, Toronto, Ontario, Canada, 10 Institute for Clinical Evaluative Sciences, Toronto, Ontario, Canada

☯ These authors contributed equally to this work.
* sheliza.halani@mail.utoronto.ca

**Data Availability Statement:** All relevant data are within the manuscript and its Supporting Information files.

## Abstract

### Background

Zika virus (ZIKV) has generated global interest in the last five years mostly due to its resurgence in the Americas between 2015 and 2016. It was previously thought to be a self-limiting infection causing febrile illness in less than one quarter of those infected. However, a rise in birth defects amongst children born to infected pregnant women, as well as increases in neurological manifestations in adults has been demonstrated. We systemically reviewed the literature to understand clinical manifestations and health outcomes in adults globally.

### Methods

This review was registered prospectively with PROPSERO (CRD 42018096558). We systematically searched for studies in six databases from inception to the end of September 2020. There were no language restrictions. Critical appraisal was completed using the Joanna Briggs Institute Critical Appraisal Tools.

### Findings

We identified 73 studies globally that reported clinical outcomes in ZIKV-infected adults, of which 55 studies were from the Americas. For further analysis, we considered studies that met 70% of critical appraisal criteria and described subjects with confirmed ZIKV. The most common symptoms included: exanthema (5,456/6,129; 89%), arthralgia (3,809/6,093; 63%), fever (3,787/6,124; 62%), conjunctivitis (2,738/3,283; 45%), myalgia (2,498/5,192;

**Funding:** Funding was provided by the Canadian Institutes of Health Research—Team grant—FRN149784. (https://cihr-irsc.gc.ca/e/193.html) This research was supported, in part, by a Canada Research Chair in Economics of Infectious Diseases held by BS (CRC-950-232429, https://www.chairs-chaires.gc.ca/home-accueil-eng.aspx) The funders had no role in study design, data collection and analysis, decision to publish, or preparation of the manuscript."

**Competing interests:** The authors have declared that no competing interests exist.

48%), headache (2,165/4,722; 46%), and diarrhea (337/2,622; 13%). 36/14,335 (0.3%) of infected cases developed neurologic sequelae, of which 75% were Guillain-Barré Syndrome (GBS). Several subjects reported recovery from peak of neurological complications, though some endured chronic disability. Mortality was rare (0.1%) and hospitalization (11%) was often associated with co-morbidities or GBS.

## Conclusions

The ZIKV literature in adults was predominantly from the Americas. The most common systemic symptoms were exanthema, fever, arthralgia, and conjunctivitis; GBS was the most prevalent neurological complication. Future ZIKV studies are warranted with standardization of testing and case definitions, consistent co-infection testing, reporting of laboratory abnormalities, separation of adult and pediatric outcomes, and assessing for causation between ZIKV and neurological sequelae.

### Author summary

Interest in Zika virus (ZIKV) has increased in the last decade due to its emergence and rapid spread in the Americas. In this review, we examine ZIKV clinical manifestations and sequelae in adults. Among studies reporting subjects with confirmed ZIKV and critical appraisal scores of at least 70%, symptoms reported include exanthema, fever, arthralgia, conjunctivitis, myalgia, headache, and diarrhea. Neurological sequelae in this group occurred in 0.3% of subjects, of which 75% were Guillain-Barré Syndrome (GBS). Recovery from GBS was variable: some patients returned to health and others endured chronic disability. Mortality was rare (0.1%). Hospitalization (11%) was often associated co-morbidities or GBS; this percentage perhaps reflects studies in which all reported subjects were hospitalized. Synthesizing reported data is challenging given the wide range of case definitions and ZIKV testing practices.

## Introduction

Zika virus (ZIKV) was first identified in sentinel rhesus macaque monkeys in 1947 in Uganda, with the first report of human disease in 1952[1,2]. The virus has two dominant lineages, historically found in Africa and South-East Asia [3,4]. ZIKV is a single-stranded RNA virus, belonging to the *flavivirus* genus, which is a part of the *Flaviviridae* family of viruses [5–7]. There is overlap in terms of epidemiology and transmission cycles between ZIKV and other vector-borne diseases, in particular dengue and chikungunya [8]. Most *Aedes* mosquitoes are capable of carrying and transferring ZIKV, with *Aedes aegypti* and *Aedes albopictus* being recognized as the main vectors in human transmission [2,7]. Pregnancy, blood transfusions and sexual transmission are confirmed as other routes of transmission [3,9–14].

ZIKV has generated substantial global interest in the last five years mostly due to its recent re-emergence and rapid spread in the Americas between 2015 and 2016[15–17]. Prior to 2015 no infections were reported in the Americas [18]. Previously thought to be a self-limiting infection causing febrile illness in 20% of those infected, new concerns have arisen due the sharp rise in birth defects amongst children born to infected pregnant women [19–21] ZIKV has also been associated with long-term neurological sequelae in adults [22]. This systematic

review synthesizes the existing literature on clinical manifestations and sequelae of ZIKV infection specifically in adults. Knowledge generated from this review will aid in informing when to test for ZIKV and will provide information regarding the risk of clinical outcomes and prognosis with ZIKV infection in adults.

## Methods

### Protocol and registration

We report this systematic review in accordance with the Preferred Reporting Items for Systematic Reviews and Meta-Analyses (PRISMA) statement [23]. We registered the study protocol on PROSPERO, a database of registered systematic reviews (Registration number: CRD42018096558, https://www.crd.york.ac.uk/prospero/display_record.php?RecordID=96558) in May 2018.

### Information sources

We systematically searched for relevant studies in MEDLINE (Ovid), Embase (Ovid), PubMed, CINAHL (EBSCO), LILACS (Literatura Latino-americana e do Caribe em Ciências da Saúde) and WHO's ICTRP clinical trials registries database.

### Search strategy

An information specialist with expertise in systematic reviews designed and carried out the search following the methodology of a Cochrane systematic review [24,25]. Our broad search was composed of "Zika Virus" or "Zika Virus Infection" controlled vocabulary (MeSH) and corresponding natural language terms. There were no language restrictions. We initially searched for studies published from inception through to April 2018 and then updated the search to include studies until September 15 2020. Full search strategy can be accessed in S1 Text.

### Eligibility criteria

The following studies were included: observational studies (cross-sectional, case-control and cohort studies), indexed reports, and case reports and case series reporting with at least 10 participants that reported on health outcomes for adults ($\geq$18 years). Randomized controlled trials (RCT) investigating the outcomes of interest were also included; those that focused on treatment safety and efficacy were not. References reporting on the effects of ZIKV in fetuses, and not the pregnant mother, were excluded from the review, as were any studies focusing on children (<18 years old). Studies that reported on adult outcomes but also included children, and in which the data could not be separated, were included, while stating the mixed population in the Results. We excluded publications such as editorials, letters and news articles, and animal studies. Abstracts and conference proceedings were excluded.

### Study selection, data collection, synthesis

Two reviewers independently screened titles and abstracts of identified records followed by reviewing the full text against inclusion/exclusion criteria using DistillerSR (Evidence Partners, Ottawa, Canada). Discrepancies were resolved through consensus or consultation with a third reviewer or the larger research team.

Data was then extracted on a pre-designed, pilot-tested data extraction form on Microsoft Excel including study characteristics, subject demographics characteristics, ZIKV signs and symptoms, and clinical outcomes, including descriptive statistics and measures of association.

Study type was also determined by the two reviewers and classification was performed in keeping with literature by Dekker and colleagues and reference textbook by Fletcher and colleagues [26,27]. Discrepancies in data extraction were resolved through consensus or consultation with a third reviewer.

We descriptively summarized the results using frequencies, percentages, and ranges. We did not perform a meta-analysis given the heterogeneity of the studies included.

### Risk of bias assessment

We critically appraised included studies using the Joanna Briggs Institute Critical Appraisal Tools [28]. Each reviewer scored the studies and a third reviewer was consulted in the case of disagreement. In assessing whether outcomes were measured in a standard, valid, and reliable way, standard criteria were used as a benchmark; for example the Brighton criteria for Guillain Barré Syndrome (GBS) [29,30]. To calculate the total percent criteria met, we removed criteria that were found to be "not applicable" from the denominator.

## Results

Here we outlined the results of our study selection, study characteristics, subject demographics, risk of bias assessment, data on case definitions, health outcomes and further analyses (travel-associated cases, co-infections, comorbidities and pre-existing conditions, and laboratory manifestations).

### Results of study selection

Through the database search, 15,956 articles were identified and 12,068 deduplicated titles and abstracts were screened against eligibility criteria, of which 837 were selected for full-text review. Of these 837 articles, 499 pertained to pregnant women and/or children only and were excluded. Another 265 studies were excluded mainly because they were case reports or case series with fewer than ten participants, conference abstracts, or letters to the editor. Seventy-three studies reported on adult populations and were included in the review. (Fig 1)

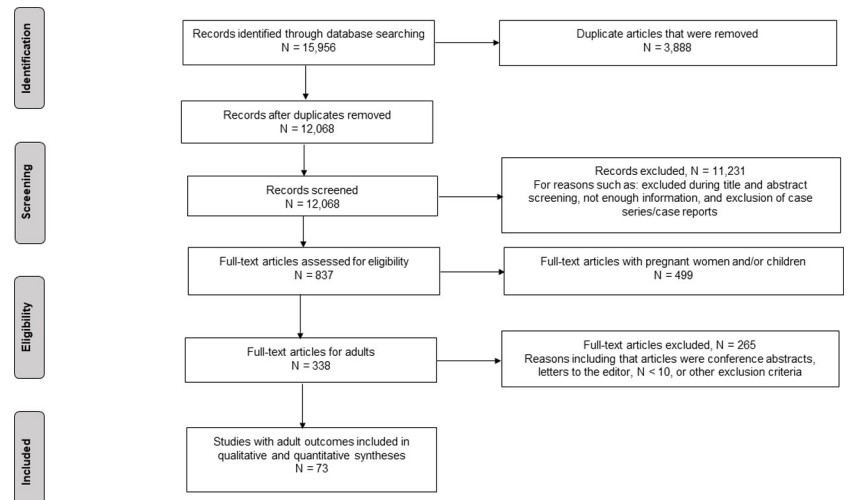

**Fig 1. PRISMA Flow Diagram.** PRISMA Flow Diagram illustrating identification, screening, eligibility, and inclusion of articles related to adult health outcomes with ZIKV.

## Study characteristics

Of the 73 included studies, there were eight (11%) case-control studies, 13 (18%) case series, 13 (18%) cohort studies, and 39 (53%) cross-sectional studies. Forty one studies were from the Latin America and the Caribbean, 14 from the USA and Canada, eight from Europe, four from Asia, three from Oceania, two from India, and one from the Middle East. Forty four studies were conducted using public health reporting and/or surveillance data, while others collected data from hospitals, healthcare centers, travel centers, or clinics. (S1 Fig and S1 Table)

From all studies, the total number of ZIKV-infected patients reported was 309,649 including suspected, probable, and confirmed cases. Sample size ranges and medians were: case-control studies had a median of 46 subjects (range: 18 to 6,117 subjects), case series had a median of 20 subjects (range: 10 to 37,878 subjects), cohort studies had a median of 101 subjects (range: 17 to 7,722 subjects), and cross-sectional studies had a median of 948 subjects (range: 34 to 108,087 subjects).

## Subject demographics

Of the 49 studies that reported the median age for all subjects in the study, the median age ranged from 20 to 61 years. From 61 studies in which the number of female subjects with ZIKV were reported, there were 114,800 females out of 188,391 infected subjects (61%). Twenty one studies reported cases of ZIKV associated with travel, and 37 studies included pregnant women in their study subjects. (Table 1)

## Risk of bias in individual studies

Sixty-three studies met at least 70% of critical appraisal criteria (all eight case-control studies, 10 of 13 case series, 11 of 13 cohort studies, and 34 of 39 cross-sectional studies) when assessed using the Joanna Briggs Institute (JBI) critical appraisal tool. Further details on how the critical appraisal tool was utilized and study classification can be found in S3 Text.

## ZIKV case definitions

Clinical case definitions of ZIKV infection varied widely between studies, and are listed in S5. Many definitions required fever and/or rash for inclusion, such as the widely used Pan-American Health Organization (PAHO) definition. Sometimes, other cardinal symptoms associated with ZIKV infection (e.g., conjunctivitis, arthralgias) qualified as independent inclusion criteria. Number of symptoms required for eligibility ranged from 1–4. Other studies included patients based on a complication of ZIKV infection, most commonly neurological sequelae or specifically GBS.

Forty of 73 (55%) studies included some form of confirmatory testing (in any or all subjects) that met at least one criterion of the World Health Organization (WHO) criteria for 'confirmed' ZIKV infection [104]. The two main methods were RT-PCR of serum and/or urine, and positive serum ZIKV IgM with positive viral neutralization testing (Table A in S5 Text)

In the percentages calculated in the subsequent section on health outcomes (and in Table 2), we included only studies that met at least 70% of the critical appraisal criteria. We reported on a subset of subjects with 'confirmed' ZIKV infection either by strict WHO definition or similar to this definition as assessed by the authors of the current systematic review. Studies in which the definition of ZIKV infection was similar to the WHO definition of 'probable' or 'suspected' ZIKV were included separately (Table 3). S4 Text and S2 Fig outline more detail to the approach taken to present the data based on diagnosis or type of testing for ZIKV.

**Table 1. Study Characteristics of Seventy-Three Adult Studies with ZIKV-Infected Cases Organized by Study Type.**

| | Location | Total Number of Subjects, n | Total Female Subjects, n (%) | Age, in years (definition) | Total Number of ZIKV Infections, n | Female Subjects with Infection, n (%) | Travel-Associated Cases? | Pregnant Women, n (%) | Confirmatory ZIKV Testing?* | Critical Appraisal, % |
|---|---|---|---|---|---|---|---|---|---|---|
| **Case-Control Studies** | | | | | | | | | | |
| Anaya, 2017 [31] | Cúcuta, Colombia[a] | 6117 | 4382 (71.2) | 28 (median) | 6117 | 4382 (71.2) | | 1936 (44.2) | Yes | 100 |
| Cao-Lormeau, 2016[32] | Tahiti, French Polynesia | 210 | 11 (26) | 42 (median) | 41 | 11 (26) | | | Yes | 90 |
| Geurts vanKessel, 2018[33] | Bangladesh | 418 | 152 (36.4) | 27 (median) | 18 | | | | Yes | 100 |
| Salinas, 2017 [34] | Barranquilla, Colombia | 40 | 19 (48) | 47 (median) | 10 | | | | No | 90 |
| Styczynski, 2017[35] | Salvador metropolitan area, Brazil | 41 | 19 (46) | 44 (median) | 21 | | | | No | 100 |
| Gongora-Rivera, 2020 [36] | Northeastern Mexico | 50 | 19 (38) | 40.5 (median) | 14 | -- | -- | -- | Yes | 88.9 |
| Rivera-Correa, 2019[37] | Salvador, Brazil | 18 | -- | -- | 15 | -- | -- | -- | Yes | 90 |
| Kozak, 2020 [38] | Ontario, Canada | 60 | 32 (53) | 52.5 (median, DENV coinfection), 44 (non DENV) | 60 | 32 (53) | Yes | NA | Yes | 100 |
| **Case Series** | | | | | | | | | | |
| Acevedo, 2017 [39] | Guayaquil, Ecuador | 16 | 9 (47) | 42.1 (mean) | 9 | 3 | | | Yes | 55.6 |
| Arias, 2017 [40] | Cúcuta, Colombia[b] | 19 | 7 (37) | 44 (mean) | 19 | 7 (37) | | | Yes | 80 |
| Baskar, 2018 [41] | Pondicherry, India | 90 | 32 (35.6) | 30–40 (a third of patients in this age range) | 14 | 5 (36) | | | Yes | 80 |
| Chang, 2018 [42] | Northern Colombia | 19 | 7 (37) | 50 (median) | 19 | 7 (37) | | 0 | Yes | 70 |
| Dirlikov, 2018 [43] | Puerto Rico | 123 | 55 (45) | 55 (median) | 71 (of 107 sent for arboviral testing) | 37 (52.1) | | | Yes | 90 |
| Duijster, 2016 [44] | The Netherlands | 18 | 12 (67) | 54 (median) | 18 | 12 (67) | Yes | 1 (6) | Yes | 55.6 |
| Lynch, 2019 [45] | Baranquilla, Colombia | 17 | 12 (71) | 49 (median) | 17 | 12 | -- | -- | No | 70 |
| Sebastián, 2017[46] | Eight Latin American countries[c] | 10 | 4 (40) | 42 (mean) | 10 | 4 (40) | | | Yes | 66.7 |
| Uncini, 2018 [47] | Cúcuta, Colombia | 20 | 13 (65) | 42 (median) | 20 | 13 (65) | | | No | 70 |
| Van Dyne, 2019[48] | Puerto Rico | 37878 (47 had ZIKV-associated TCP) | 22 of 47 (47) | 39.5 (median in severe TCP), 49 (median in non-severe TCP) | 37878 | | | | Yes | 80 |
| Watrin, 2016 [49] | Tahiti, French Polynesia | 42 | 11 (26) | 42 (median) | 36 | | | | No | 100 |

(Continued)

**Table 1.** (Continued)

| | Location | Total Number of Subjects, n | Total Female Subjects, n (%) | Age, in years (definition) | Total Number of ZIKV Infections, n | Female Subjects with Infection, n (%) | Travel-Associated Cases? | Pregnant Women, n (%) | Confirmatory ZIKV Testing?* | Critical Appraisal, % |
|---|---|---|---|---|---|---|---|---|---|---|
| Chaumont, 2020[50] | Guadelope | 171 | 78 (45.6) | 49 (median) | 23 (21 adults, 2 children) | 13 (56.5) | NR | NR | Yes | 100 |
| Lannuzel, 2019[51] | French West Indes 2016 outbreak, Guadelope and Martinique | 87 | 43 (49.4) | 54 (median)^l | 87 | 43 (49.4) | NR | NR | Yes | 100 |
| **Cohort Studies** | | | | | | | | | | |
| Calvet, 2018 [52] | Rio de Janeiro, Brazil | 101 | 42 (42) | 41.8 (median) | 77 | 37 | | | Yes | 80 |
| da Silva, 2017 [53] | Rio de Janeiro, Brazil | 40 | 15 (38) | 44 (median) | 35 | 13 (37) | | | Yes | 88.9 |
| de Laval, 2018 [54] | French Guiana, northeast South America | 49 | 10 (20) | 38 (mean) | 49 | 10 (20) | | | Yes | 88.9 |
| Kam, 2017[55] | Campinas, Brazil | 95 | 66 (69) | 35 (median) | 95 | 66 (69) | | 6 (6) | Yes | 57.1 |
| Lozier, 2018 [56] | Puerto Rico | 367 | 215 (59) | 59.5 (median ZIKV +), 58 (median, ZIKV-) | 114 | 63 (55) | | 2 (0.5) | Yes | 80 |
| Meltzer, 2019 [57] | Israeli travelers^a | 1,188 | 641 (54) | 29.9 (mean) | 30 | 15 (50) | Yes | 388 pregnant or spouse pregnant | Yes | 90 |
| Ng, 2018[58] | Singapore | 40 | 16 (40) | 34 (median) | 40 | 16 (40) | Yes | 0 | Yes | 100 |
| Sokal, 2016 [59] | Paris, France | 17 | 10 (59) | 42 (mean) | 17 | 10 (59) | Yes | 1 (6) | Yes | 66.7 |
| Vega, 2018 [60] | Santa Luzia, Brazil | 7,063 | 2009 (57) | 29.2 (median) | 12 | (100) | NR | 10 | Yes | 100 |
| Petridou, 2019 [61] | United Kingdom | 7,722 | (56) | NR | 374 (499 positive testing) | (55) of confirmed cases | Yes | 16 (0.002) | Yes | 71.4 |
| Hunsberger, 2020[62] | Southern Mexico | 366 | 221 (60) | 33.7 (median) for Zika^m | 33 | 20 (61) | NA | NA | Yes | 88.9 |
| Crespillo-Andújar, 2019 [63] | Madrid, Spain | 817 | 459 (56) | 36 (median) | 51 | 28 (60 of symptomatic) | Yes | 2 | Yes | 80 |
| El Sahly, 2018 [64] | United States | 56 | 40 | 44 (median, cases) 31 (controls) | 45 | 31 (68.9) | Yes | NA | Yes | 90 |
| **Cross-sectional Studies** | | | | | | | | | | |
| Adams, 2016 [65] | Puerto Rico, United States of America | 16522 | . | | 5351 | | | 9343 (57); 672 confirmed/ presumed ZIKV | Yes | 66.7 |
| Armstrong, 2016[66] | United States of America^d | 115 total (104 adults) | 75 (65) | 38 (median) | 115 | 75 (65) | Yes | | Yes | 80 |
| Azeredo, 2018 [67] | Campo Grande, Brazil | 134 | | 31 (median in ZIKV positive subjects) | 38 | 21 (55) | No | 5 (13) | Yes | 100 |
| Boggild, 2017 [68] | Canada | 1118 | | 36 (median in ZIKV positive subjects) | 41 | 24 (59) | Yes | 3 of 41 (7) | Yes | 87.5 |

(Continued)

Table 1. (Continued)

| | Location | Total Number of Subjects, n | Total Female Subjects, n (%) | Age, in years (definition) | Total Number of ZIKV Infections, n | Female Subjects with Infection, n (%) | Travel-Associated Cases? | Pregnant Women, n (%) | Confirmatory ZIKV Testing?* | Critical Appraisal, % |
|---|---|---|---|---|---|---|---|---|---|---|
| Brasil, 2016 [69] | Rio de Janeiro, Brazil[e] | 364 | 158 of 262 tested for ZIKV (60.3) | 37 (median) in ZIKV-tested | 364 | 158 of 262 tested for ZIKV (60.3) | No | 4 of 119 confirmed ZIKV (3) | Yes | 83.3 |
| Brenciaglia, 2018[70] | Grenada[f] | 514 | 380 of 511 (74) | 30 (median); 73 patients under age 20 | 207 | 148 (72) | | 117 of 380 (31) | Yes | 87.5 |
| da Silva Brito, 2018[71] | Rio de Janeiro | 113 | 71 (63) | Fourth decade of life most affected (21.2%) | 113 | 71 (63) | | | No | 33.3 |
| Daudens-Vaysse, 2016 [72] | Martinique, French Territories of America[g] | 9077 | :: | 43 (mean age of confirmed cases) | 9,077 | 142 of 203 confirmed cases (70) | No | 44 of 7600 | Yes | 71.4 |
| Duffy, 2009 [73] | Yap State, Federated States of Micronesia | 185 | 66 of 108 confirmed/probable (61) | 36 (median of confirmed/probable cases) | 185 | 66 of 108 confirmed/probable (61) | No | . | Yes | 83.3 |
| Francis, 2018 [74] | Caribbean Public Health Agency (CARPHA) member states (CMS)[a] | 5614 | 1200 of 1447 confirmed ZIKV infection (83) | 30 (median among confirmed) | 5614 | 1200 of 1447 confirmed ZIKV infection (83) | | 614 of 1200 (51) | Yes | 87.5 |
| Hall, 2018[75] | United States of America[a] | 5168 (4118 above age 20 years) | 3310 (64) | 37 (median) | 5168 (4118 above age 20 years) | 3310 (64) | Yes | 469 (14) | Yes | 66.7 |
| Hamer, 2017 [76] | Americas (South America, Central America including Mexico and Caribbean)[a] | 93 (85 subjects above age 20 years) | 58 (62) | 41 (median) | 93 (85 subjects above age 20 years) | 58 (62) | Yes | 4 (4) | Yes | 71.4 |
| Ho, 2017[77] | Singapore[h] | 455 | 192 (42) | 36 (median) | 455 | 192 (42) | | 17 (4) | Yes | 71.4 |
| Huits, 2019 [78] | Belgium | 462 | 235 (47) | 32 (median); 38 (median of ZIKV cases) | 49 | 27 (55) | Yes | 59 of 462 (13) pregnant/ partner pregnant | Yes | 85.7 |
| Jimenez Corona, 2016 [79] | Mexico[a] | 93 | 61 (66) | 35 (mean) | 93 | 61 (66) | No | 8 of 93 (9) | Yes | 80 |
| Journel, 2017 [80] | Haiti[i] | 3036 | ~56% of confirmed cases | 34 (median age of 19 confirmed cases) | 3036 | ~56% of confirmed cases | Yes | 22 of 3036 (0.7) | Yes | 57.1 |
| Lee, 2016[81] | New York, United States of America | 3605 | | | 182 | | Yes | 20 (11) | Yes | 75 |
| Malta, 2017 [82] | Salvador metropolitan area, Brazil | 138 | 25 of 57 with neurological manifestations | 44 (median age) of those with neurological manifestations | 30 | | | | Yes | 100 |
| McGibbon, 2018[83] | New York City, United States of America | 1080 noncongenital cases (1102 total) | 864 of 1080 noncongenital ZIKV cases (80) | 33 (median age of 1080 cases) | 1080 noncongenital | 864 (80) | Yes | 412 (38) | Yes | 83.3 |

(Continued)

**Table 1.** (Continued)

| | Location | Total Number of Subjects, n | Total Female Subjects, n (%) | Age, in years (definition) | Total Number of ZIKV Infections, n | Female Subjects with Infection, n (%) | Travel-Associated Cases? | Pregnant Women, n (%) | Confirmatory ZIKV Testing?* | Critical Appraisal, % |
|---|---|---|---|---|---|---|---|---|---|---|
| Méndez, 2017 [84] | Colombia[j] | 108,087 | 70,478 of 106,455 (66) | Highest attack rate in age 25 to 29 years (375 per 100,000 population) | 108,087 | 70,478 of 106,455 (66) | | 19,963 (18.5) | Yes | 71.4 |
| Millet, 2017 [85] | Barcelona, Spain | 118 | | 35 (median of the 44 confirmed cases in Barcelona) | 118 cases notified (75 lab-confirmed) | 25 of 44 confirmed cases in Barcelona (57) | | 6 of 44 | Yes | 71.4 |
| Parra, 2016 [86] | Colombia | 58,790 | 30 of 68 patients with GBS (44) | 47 (median age of 68 patients with GBS) | 58,790 | 30 of 68 patients with GBS (44) | | | Yes | 100 |
| Rozé, 2017 [87] | Martinique, French West Indies | 34 | | 61 (median age of 23 recent ZIKV cases) | 27 (23 recent ZIKV infection) | 8 of 23 (35) | | | Yes | 100 |
| Ryan, 2017 [88] | Commonwealth of Dominica[a] | 1263 | 863 of 1255 (69) | 27 (median for 1245 that reported age) | 1263 | 863 of 1255 (69) | | 16 of 54 women that reported (30) | Yes | 60 |
| Schirmer, 2018 [89] | United States of America | 1538 | | 58.7 (mean) | 736 | 81 (11) | Yes | 4 of 81 (5) | Yes | 100 |
| Thomas, 2016 [90] | Puerto Rico, United States of America | 155 | 18 of 30 confirmed cases (60) | 40 (median age of confirmed cases) | 155 | 18 of 30 confirmed cases (60) | Yes | 1 of 30 (3) | Yes | 83.3 |
| Vroon, 2017 [91] | Paramaribo, Suriname | 102 | 64 (63) | 46 (median age) | 77 | 48 of 77 (62) | | | Yes | 100 |
| Webster-Kerr, 2017 [92] | Jamaica[k] | 5426 | | | 5426 | | | 604 (11) | Yes | 100 |
| Grajales-Muniz, 2019 [93] | Mexico | 43,725 | 27,832 (63.7) | 30[n] | 43,725 | 27,832 (63.7) | Yes | 1,082 confirmed positive and pregnant (4,168 of total) | Yes | 100 |
| Valle, 2019 [94] | Atlanta, USA | 46 | 28 (60.1) | 34 (median, cases), 33.5 (noncases) | 8 | 3 | Yes | 0 | Yes | 100 |
| Martinez, 2019 [95] | Spain | 512 | 327 (63.9) | 34 (median) | 507 | 327 (63.9) | Yes | 86 | Yes | 75 |
| Silva, 2019 [96] | Brazil | 948 | 390 (41) | 20 (median) | 14 | 7 (50) | NA | NA | Yes | 87.5 |
| Mercado-Reyes, 2019 [97] | Colombia | 23,871 | NA | NA | 10,118 | 25 (76.5 of 34 with co-infection) | NA | 14 (41.2 of 34 with coinfection) | Yes | 100 |
| Garcell, 2020 [98] | La Habana, Cuba | 1,541 | 983 (63.8) | 43 (mean) | 279 | 163 (58.4) | NA | NA | Yes | 85.7 |
| Del Carpio-Orantes, 2020 [99] | Mexico | 10,327 | 4,655 (45.1) | NR | 3,529 | 1,154 (32.7) | NR | 275 | Yes | 85.7 |
| Castañeda-Martinez, 2020 [100] | Michoacán, Mexico | 700[o] | 478 (68.2) | 30.95 | 700 | 478 (68.2) | NR | 137 | Yes | 83.3 |

*(Continued)*

**Table 1.** (Continued)

| | Location | Total Number of Subjects, n | Total Female Subjects, n (%) | Age, in years (definition) | Total Number of ZIKV Infections, n | Female Subjects with Infection, n (%) | Travel-Associated Cases? | Pregnant Women, n (%) | Confirmatory ZIKV Testing?* | Critical Appraisal, % |
|---|---|---|---|---|---|---|---|---|---|---|
| Sharma, 2019 [101] | Rajasthan, India | 1,925 | NA | 27.5 (mean)[p] | 111 | 59 (53) | NA | 27 (2.5) | Yes | 83.3 |
| Vazquez, 2019 [102] | Paraguay | 580 | 329 (56.7) | 24 (median) | 45 | 28 (62.2) | NA | NA | Yes | 100 |
| Phan, 2019 [103] | Southern Vietnam | 2,190 | 1,348 | NR | 214 | 147 (68.7) | NR | 47 | Yes | 83.3 |

* Note: Number of infections includes all confirmed probable and suspected cases (in the situations where the primary paper divided these)

This question refers to whether the primary study had *any* confirmatory testing in their methods exactly by World Health Organization (WHO) criteria. [104]

[a] Total numbers in these studies include children. [57,75,76,88,105–107]

[b] In this study, there was one subject that had ZIKV infection confirmed by RT-PCR (out of 19 cases). Since this study used the Instituto Nacional de Salud (INS) "confirmed cases by clinical criteria" then all 19 cases are discussed in Table 3. [40]

[c] In the Sebastián et al. (2017) study, the eight countries enrolled were Colombia, Venezuela, El Salvador, Guatemala, Puerto Rico, Ecuador, Peru, and Chile. [46]

[d] Total number was 116 but have removed one infant from this table, therefore this study reported on 115 adults infected with ZIKV. [66]

[e] The study by Brasil et al. in 2016 includes children. Among the 119 confirmed ZIKV cases, 115 were above the age of 15. [69]

[f] Authors made comparison to geographic areas among the country and different parishes. The study focused on investigating the cases in Grenada and this is reflected in the tables in this manuscript. There were also children in this study from range of one day old to 90 years old (age range). [70]

[g] Daudens-Vaysse et al. reported on 9077 suspected cases in the French Territories of America and 7,600 of these were from Martinique; 58 from Saint-Martin, 1,030 from French Guiana, and 389 from Guadeloupe. Total confirmed among the territories was 249 cases. [77]

[h] Total numbers in this study included 25 children. [77]

[i] This study included children and congenital microcephaly cases; 3036 suspected cases included adults/children (2972), pregnant women (22), GBS (13), and congenital microcephaly (29). Nineteen confirmed cases were adults/children (17) and two pregnant women. [80]

[j] Méndez et al. study–of the total 108,087 Zika virus disease cases, this included 9,963 pregnant women, 710 associated with microcephaly, and 453 ZVD-associated to GBS. Of the 9802 confirmed cases, included 6365 pregnant women and 174 cases of microcephaly. [84]

[k] From total study, 5426 met case definitions but 91 laboratory-confirmed. They examined in more detail, epidemiological weeks 1–30, in which there were 4648 cases of ZIKV (4567 suspected and 72 confirmed).[92]

[l] Includes 6 children equal to or less than 16 years

[m] Includes participants 12 years and older

[n] 3.5% of RT-PCR confirmed ZIKV cases were in the age range of 0 to 14 years, and 55.2% of RT-PCR confirmed ZIKV cases the age range of 15 to 29 years.

[o] Includes 134 children equal to or less than 14 years

[p] Two percent of cases were in the 0 to 10 year age range, 39% were in the 11 to 20 year ago range.

AIDP = Acute Inflammatory Demyelinating Polyneuropathy, AMAN = Acute motor axonal neuropathy, CSF = Cerebrospinal fluid, DENV = Dengue virus, ELISA = Enzyme-Linked Immunosorbent Assay, IFI = indirect immunofluorescence

GBS = Guillain Barre Syndrome, PRNT = Plaque reduction neutralization test; RT-PCR = Reverse transcriptase-polymerase chain reaction

RVP = Reporter Virus Particles, SIRS = systemic inflammatory response syndrome

TCP = Thrombocytopenia, URTI = Upper Respiratory Tract Infection, VNT = Virus neutralization test, ZCD = Zika Chikungunya Dengue, ZIKV = Zika virus

**Table 2. Symptomatology, Neurological Complications, Hospitalization and Mortality in ZIKV-Confirmed Patients with Laboratory-Confirmation Definitions Similar to World Health Organization.**

| | Total Subjects with Confirmed ZIKV Infection, n* | Symptoms: Symptom Denominator, n (definition)** | Fever, n | Exanthema, n | Conjunctivitis, n | Myalgia, n | Arthralgia, n | Diarrhea, n | Headache, n | Total Subjects with Neurologic Sequelae, n | Guillain Barré Syndrome, n | Hospitalization, n | Death, n |
|---|---|---|---|---|---|---|---|---|---|---|---|---|---|
| Anaya, 2017[31] | 655 | -- | -- | -- | -- | -- | -- | -- | -- | -- | -- | -- | -- |
| Cao-Lormeau, 2016[32] | 41 | Variable denominator | 18 of 31 | 29 of 36 | 15 of 31 | | 23 of 31 | | | 41 | 41 | 41 | 0 |
| Geurts vanKessel, 2018[109] | 18 | 9 (ZIKV VNT-positive) | | | | | | 2 | | 18 | 18 | 18 | 2 |
| Chang, 2018[42] | 19 | Variable | 12 of 18 | 12 of 18 | 6 of 18 | 13 of 18 | 13 of 18 | 5 of 18 | 9 of 15 | 19 | 19 | | 0 |
| Lynch, 2019[45] | 17 | 16 | 15 | 12 | 8 | 14 | 15 | 6 | | 17 | 17 | | 1 of 8 |
| Uncini, 2018[47] | 20 | 20 | 15 | 16 | 12 | | 15 | | | 20 | 20 | 20 | |
| Calvet, 2018[52] | 43 | 43 (RT-PCR) | 21 | 40 | 28 | 30 | 27 | 12 | 24 | | | 0 | 0 |
| da Silva, 2017[53] | 35 | 35 | 31 | 30 | 10 | 23 | 15 | | 11 | 35 | 27 | 35 | 2 |
| de Laval, 2018[54] | 49 | 49 | 26 | 45 | 28 | 25 | 26 | 13 | 34 | 0 | 0 | | 0 |
| Ng, 2018[58] | 40 | 40 | 32 | 40 | 24 | | 15 | 6 | 10 | 0 | 0 | 40 | |
| Azeredo, 2018[67] | 38 | 15 | 13 | 12 | 8 | 12 | 10 | 3 | 13 | 0 | 0 | 2 DENV/ZIKV patients hospitalized | |
| Boggild, 2017[68] | 41 | 41 | 33 | 36 | 5 | 19 | 22 | 5 | 17 | 3 | 2 | (all sought medical care) | |
| Brasil, 2016[69] | 119 | 119 | 43 | 115 | 66 | 73 | 75 | 23 | 78 | 1 (of total 262 cases) | | 1 (of total 262 cases) | 0 |
| Daudens-Vaysse, 2016[72] | 500 | 500 | 335 | 446 | 255 | 252 | 328 | | 28 of 203 | 6 | 6 | 1 | 1 |
| Duffy, 2009[73] | 49 | 31 (reported symptoms) | 20 | 28 | 17 | 15 | 20 | | 14 | | | | |
| Francis, 2018[74] | 1447 | 1289 adults | 791 | 1111 | 431 | 138 | 809 | | 176 | 13 (of 1447) | 13 (of 1447) | 0 | 4 (of 1447) |
| Ho, 2017[77] | 455 | 149 (Adults with symptoms/test outcomes reported) | 118 | 139 | 35 | 63 | 34 | .. | 35 | .. | 0 | 149 | .. |
| Huits, 2019[78] | 49 | 46 | 26 | 43 | 11 | 12 | 21 | 7 | 3 | | 0 | | |
| Jimenez Corona, 2016[79] | 93 | 93 | 90 | 87 | 83 | | 78 | | | 2 | 0 | 2 | 0 |
| Lee, 2016[81] | 182 | | | | | | | | | 2 | 2 | | |
| McGibbon, 2018[83] | 725 | 725 | 399 | 607 | 321 | | 448 | | | 2 | 2 | | |
| Millet, 2017[85] | 75 | 44 | 27 | 38 | | 13 | 26 | | 18 | | 0 | 0 | |
| Vroon, 2017[91] | 21 | 21 | 9 | 4 | 3 | 15 | 9 | .. | 12 | .. | .. | 8 | 1 |

(Continued)

**Table 2.** (Continued)

| | Total Subjects with Confirmed ZIKV Infection, n* | Symptoms: Symptom Denominator, n (definition)** | Fever, n | Exanthema, n | Conjunctivitis, n | Myalgia, n | Arthralgia, n | Diarrhea, n | Headache, n | Total Subjects with Neurologic Sequelae, n | Guillain Barré Syndrome, n | Hospitalization, n | Death, n |
|---|---|---|---|---|---|---|---|---|---|---|---|---|---|
| Webster-Kerr, 2017[92] | 91 | 72 (confirmed within EW 1–30) | 38 | 63 | 13 | 8 | 27 | | 17 | 17 suspected GBS (unknown if in ZIKV confirmed or suspected group) | 17 suspected GBS (unknown if in ZIKV confirmed or suspected group) | | |
| Gongora_Rivera, 2020[36] | 11 | 11 | -- | -- | -- | -- | -- | -- | -- | 11 | 11 | -- | -- |
| Kozak, 2020[38] | 60 | 60 | 34 | 56 | 14 | 15 | 28 | 4 | 20 | 0 | | | |
| Vega, 2018[60] | 12 | -- | 1 | 12 | -- | 2 | 1 | -- | 3 | -- | -- | -- | -- |
| Petridou, 2019[61] | 161 | -- | 113 | 150 | 18 | 52 | 72 | 8 | 38 | -- | 0 | -- | -- |
| Hunsberger, 2020 (0–7 days after symptom onset)[62] | 33 | 33 | 21 | 22 | 20 | 30 | 22 | -- | 27 | -- | -- | -- | -- |
| Hunsberger, 2020 (3–10 days after onset)[62] | 33 | 33 | | | | 19 | 14 | 2 | | 4 | | -- | -- |
| Crespillo-Andújar, 2019[63] | 26 | 25 | 22 | 23 | 8 | -- | 14 | -- | 8 | -- | -- | -- | -- |
| El Sahly, 2019[64] | 45 | 45 | 10 | 44 | 25 | 24 | 37 | -- | 24 | -- | 0 | -- | -- |
| Grajales-Muniz, 2019[93] | 1,700 | 1,700 | 1,002 | 1,642 | 1,094 | 1,232 | 1,174 | 187 | 1,287 | -- | -- | -- | 0 |
| Silva, 2019[96] | 14 | 13 | 13 | 9 | -- | 11 | 7 | -- | 12 | -- | -- | -- | -- |
| Mercado-Reyes, 2019[97] | 10,118 | 3 | 1 | 0 | -- | -- | 1 | -- | -- | 2 | -- | 26 of 34 | 3 |
| Garcell, 2020[98] | 279 | 279 | 108 | 268 | 89 | 134 | 183 | 51 | 147 | -- | -- | -- | -- |
| Del Carpio-Orantes, 2020[99] | 87 | -- | -- | -- | -- | -- | -- | -- | -- | 2 | 2 | -- | -- |
| Castañeda-Martinez, 2020[100] | 26 | 26 | 21 | 25 | 20 | 21 | 22 | 5 | 21 | -- | 0 | -- | -- |
| Sharma, 2019[101] | 111 | 111 | 91 | 32 | 18 | 72 | 62 | -- | 43 | 0 | 0 | -- | -- |
| Vasquez, 2019[102] | 45 | 45 | 44 | 10 | 2 | 31 | 30 | -- | 36 | -- | -- | 4 | -- |
| Phan, 2019[103] | 214 | 214 | 194 | 210 | 66 | 149 | 123 | -- | -- | -- | -- | -- | -- |

Note: This table reports GBS and total neurologic sequelae given that GBS was the main neurologic outcome. Please see body of text for information on other neurologic sequelae that were noted in the primary studies.

*See supplementary file S5 Text for corresponding ZIKV clinical and laboratory criteria for 'confirmed' ZIKV cases by primary authors of articles.

**The denominator for symptoms was derived from the original manuscripts.

DENV = Dengue virus, EW = Epidemiological week, GBS = Guillain-Barré Syndrome, VNT = Virus Neutralization Test, ZIKV = Zika Virus

**Table 3. Symptomatology, Neurological Complications, Hospitalization and Mortality in Cases of ZIKV Infection where by Symptoms Cannot be Separated by Form of Testing or in Probable/Suspected Cases.**

| | Suspected Cases by Authors' Definition, n* | Probable Cases by Authors' Definition, n | Confirmed Cases by Authors' Definition, n | Symptoms: Symptom Denominator, n (definition) | Fever, n | Exanthema, n | Conjunctivitis, n | Myalgia, n | Arthralgia, n | Diarrhea, n | Headache, n | Total Subjects with Neurologic Sequelae, n | GBS, n | Hospitalization, n | Death, n |
|---|---|---|---|---|---|---|---|---|---|---|---|---|---|---|---|
| Anaya, 2017 [31] | -- | 103 | -- | 102 (probable) | 72 | 86 | 59 | -- | 79 | 39 | -- | -- | -- | -- | -- |
| Salinas, 2017 [34] | 10 | 6 | | 10 | | | | | | | | 10 | 10 | | |
| Styczynski, 2017[35] | 21 (10 had evidence of recent flavivirus infection) | | | 21 | | | | | | | | 21 | 21 | 21 | |
| Arias, 2017 [40] | | | 19 | 19 | 15 | 17 | 7 | | 14 | 4 | | 19 | 19 | 19 | 0 |
| Baskar, 2018 [41] | | 14 | | 14 | | | | | | | | 14 | 14 | 14 | 1 of 8 |
| Dirlikov, 2018[43] | | 43 IgM ELISA | 28 RT-PCR | 71 | 28 | 36 | 10 | 13 | 13 | 9 | | 71 | 71 | 71 | 2 |
| Van Dyne, 2019 [48] | | | 32 (RT-PCR) | 47 ZIKV-associated thrombocytopenia (RT-PCR or IgM ELISA) | 36 | 34 | 9 | 29 | 22 | | | | | 30 admitted (40 intensive care unit) | 1 |
| Watrin, 2016 [49] | 36 | | | 36 | | | | | | | | 36 | 36 | 36 | -- |
| Calvet, 2018 [52] | 34 | | | 34 (PAHO definition of suspected ZIKV cases) | 21 | 34 | 19 | 17 | 27 | 7 | 17 | | | 0 | 0 |
| Lozier, 2018 [56] | | 79 (recent ZIKV) and 8 (recent flavivirus) | 27 (current infection) | 49 (symptomatic ZIKV positive) | 30 | 37 | 18 | 34 | 38 | 13 | 33 | -- | | (27 sought medical are) | -- |
| Meltzer, 2019 [57] | | 5 (possible) | 25 (confirmed) | | | | | | | | | | 0 | | |
| Armstrong, 2016[66] | | 87 (Serologic); two cases had serologic evidence of a recent unspecified flavivirus classified as Zika based on epidemiological link | 28 (PCR) | 115 | 94 | 113 | 43 | 63 | 76 | | | | | 4 | |
| Brasil, 2016 [69] | 364 suspected cases | | 119 | 143 suspected cases (unconfirmed) | 71 | 113 | 57 | 96 | 105 | 21 | 101 | Reported in Table 2 | | Reported in Table 2 | -- |
| Brencialgia, 2018[70] | 424 (symptomatic) | 84 (IgM) | 107 (rRT-PCR) | 191 (ZIKV-positive) | 112 | 154 | | 68 (body pain) | 97 | 26 | 74 | 8 | 8 (4 positive by IgM, 2 nonspecific anti-flavivirus IgM, 2 no evidence of ZIKV) | | |
| (same study as above) | 424 (symptomatic) | 84 (IgM) | 107 (rRT-PCR) | 233 (ZIKV-negative) | 137 | 150 | | 98 (body pain) | 144 | 28 | 92 | See above row | See above row | See above row | |
| Hamer, 2017 [76] | 16 (clinical criteria) | 13 (probable case) | 64 (confirmed case) | 93 | 71 | 82 | 37 | 56 | 67 | Number not reported | 57 | 2 | 2 | | 0 |

(Continued)

**Table 3.** (Continued)

| Study | Suspected Cases by Authors' Definition, n* | Probable Cases by Authors' Definition, n | Confirmed Cases by Authors' Definition, n | Symptoms: Symptom Denominator, n (definition) | Fever, n | Exanthema, n | Conjunctivitis, n | Myalgia, n | Arthralgia, n | Diarrhea, n | Headache, n | Total Subjects with Neurologic Sequelae, n | GBS, n | Hospitalization, n | Death, n |
|---|---|---|---|---|---|---|---|---|---|---|---|---|---|---|---|
| Huits, 2019 [78] | | | 49 (see Table 2) | 181 (non-ZIKV cases but symptomatic travelers; 14 met European CDC Clinical Case Definition) | 98 | 30 | 4 | 42 | 34 | 64 | 5 | | See Table 2 | | |
| Malta, 2017 [82] | | 30 | | 30 | | | | | | | | 30 | 25 | 30 | 1 |
| McGibbon, 2018 [110] | | 355 | 725 (see Table 2) | 355 | 61 | 98 | 38 | | 68 | | | 4 | 4 | | |
| Méndez, 2017 [84] | 108,087 total ZVD cases | | 9,802 of the 108,087 | 108,087 | | | | | | | | 453 | 453 | | |
| Parra, 2016 [86] | 33 | 18 | 17 | 68 (GBS cases) | 47 | 40 | 17 | 23 | 15 | 6 | 23 | 68 | 68 | 68 | 3 |
| Rozé, 2017 [87] | | | | 23 (recent infection) | 5 | 11 | 8 | 8 | 10 | | 8 | 23 | 23 | 23 | 2 |
| Schirmer, 2018 [89] | | 151 | 585 | Variable | 419 of 640 | 552 of 612 | 220 of 293 | 490 of 535 (arthralgia/myalgia) | 490 of 535 (arthralgia/myalgia) | | 213 of 290 | 46 | 5 | 74 | 19 |
| Thomas, 2016 [90] | 155 | | 30 | 30 | 22 | 23 | 8 | 23 | 22 | 7 | 19 | 1 | 1 | 3 | |
| Webster-Kerr, 2017 [92] | 5426 suspected | | 91 (RT-PCR) | 4576 (suspected cases in EW 1–30) | 2991 | 3238 | 1037 | 610 | 2158 | | 1499 | See Table 2 | See Table 2 | | |
| Rivera-Correa, 2019 [37] | 4 | 6 | 5 | 15 | -- | -- | -- | -- | -- | -- | -- | -- | 7 | 10 | -- |
| Chaumont, 2020 [50] | 0 | 21 | 2 | 23 | | 16 | 7 | | 15 | 2 | | 13 | | 23 | 1 |
| Lannuzel, 2019 [51] | 11 | 11 | 65 | 87 | -- | -- | -- | -- | -- | -- | -- | 87 | 38 | 77 | 3 |
| Petridou, 2019 [61] | 98 | 213 | 12 | 46^a | 22 | 34 | 5 | 19 | 19 | 1 | 6 | -- | -- | -- | -- |
| Petridou, 2019 [61] | 98 | 213 | 12 | 99^b | 67 | 69 | 3 | 28 | 57 | 5 | 12 | 0 | -- | -- | -- |
| Petridou, 2019 [61] | 98 | 213 | 12 | 68^c | 38 | 32 | 3 | 16 | 24 | 1 | 8 | 0 | -- | -- | -- |
| Petridou, 2019 [61] | 98 | 213 | 12 | 98^d | 50 | 11 | 1 | 19 | 15 | 2 | 6 | 1 | -- | -- | -- |
| Hunsberger, 2020 [62] | 366 | -- | 33 | 274^e | 238 | 112 | 132 | 244 | 249 | -- | 249 | -- | -- | -- | -- |
| Hunsberger, 2020 [62] | 366 | -- | 33 | 274^f | -- | -- | -- | 172 | 129 | 55 | -- | 52 | -- | -- | -- |
| Crespillo-Andújar, 2020 [63] | 555 | 22 | 25 | 22 (probable cases) | 14 | 12 | 1 | | 14 | | 7 | -- | -- | -- | -- |
| Crespillo-Andújar, 2020 [63] | 555 | 22 | 25 | 555 (symptomatic, negative, indeterminate or past infection) | 350 | 102 | 14 | -- | 121 | -- | 109 | -- | -- | -- | -- |
| El Sahly, 2019 [64] | 11 | 0 | 45 | 11 | 2 | 9 | 5 | 8 | 6 | -- | 8 | -- | 0 | -- | -- |
| Grajales-Muniz, 2019 [93] | 43,725 | -- | 1,700 | 42,025 | 27,450 | 40,166 | 28,481 | 33,200 | 30,415 | 5,626 | 34,100 | -- | -- | -- | 2 |

(Continued)

**Table 3.** (Continued)

| Study | Suspected Cases by Authors' Definition, n* | Probable Cases by Authors' Definition, n | Confirmed Cases by Authors' Definition, n | Symptoms: Symptom Denominator, n (definition) | Fever, n | Exanthema, n | Conjunctivitis, n | Myalgia, n | Arthralgia, n | Diarrhea, n | Headache, n | Total Subjects with Neurologic Sequelae, n | GBS, n | Hospitalization, n | Death, n |
|---|---|---|---|---|---|---|---|---|---|---|---|---|---|---|---|
| Valle, 2019 [94] | 0 | 1 | 7 | 8 | 7 | 8 | 5 | 4 (of 6) | 1 | 2 | 8 | -- | -- | -- | -- |
| Martinez, 2019 [95] | 0 | 153 | 354 | 268 | 185 | 230 | -- | -- | -- | -- | -- | 2 | 1 | 33 | 0 |
| Silva, 2019 [96] | 588 | 0 | 14 | -- | 588 | 197 | -- | 469 | 369 | -- | 524 | -- | -- | -- | -- |
| Garcell, 2020 [98] | 1,262 | 0 | 279 | 1,262 | 427 | 1,178 | 341 | 577 | 741 | 150 | 636 | -- | -- | -- | -- |
| Castañeda-Martinez, 2020 [100] | 26 | 674 | 0 | 674 | 385 | 641 | 512 | 549 | 480 | 84 | 494 | -- | 0 | -- | -- |
| Vazquez, 2019 [102] | 535 | 0 | 45 | 535 | 528 | 155 | 57 | 390 | 321 | -- | 421 | -- | -- | 75 | -- |

Note: Some studies are in both Tables 2 and 3 if the studies delineated different outcomes for different subgroups of subjects; if outcomes within a study between subjects with 'confirmed', 'probable', and 'suspected' ZIKV could not be separated, the study was reported in Table 3.

*See supplementary file S5 Text for further details on the primary authors' (of the articles in Table 3) case definitions for suspected, probable or confirmed ZIKV definitions. Of note, PRNT90 is a plaque-reduction neutralization test to detect neutralizing antibodies against a virus. One measures the titer of a subject's serum required to reduce viral plaques by 90%. [111]

[a]Seroconversion (ZIKV IgG negative to IgG positive in later sample)

[b]Probable (ZIKV IgM and IgG positive in the earliest blood sample available or ZIKV IgM strongly positive (normalised optical density ≥2.0) with no follow-up blood sample received but a very compelling clinical presentation)

[c]Likely (strongly ZIKV IgG positive (normalized optical density≥2.0) without ZIKV IgM)

[d]Doubtful (ZIKV IgM positive without ZIKV IgG seroconversion in later samples or weakly positive ZIKV serology (either IgM or IgG positive) but had a confirmed or presumptive alternative diagnosis.)

[e]0 to 7 days after symptom onset

[f]≥10 days after symptom onset

ECDC = European Center for Disease Control Clinical Case Definition = ZIKV infection defined as maculopapular rash with or without fever, and painful joints or muscles or non-purulent conjunctivitis, ELISA = Enzyme-Linked Immunosorbent Assay, EW = epidemiological week, GBS = Guillain Barré Syndrome; PAHO = Pan American Health Organization[112], RT-PCR = Reverse-Transcriptase Polymerase Chain Reaction, rRT-PCR = Real-Time Reverse Transcriptase-Polymerase Chain Reaction, ZIKV = Zika Virus, ZVD = Zika virus disease

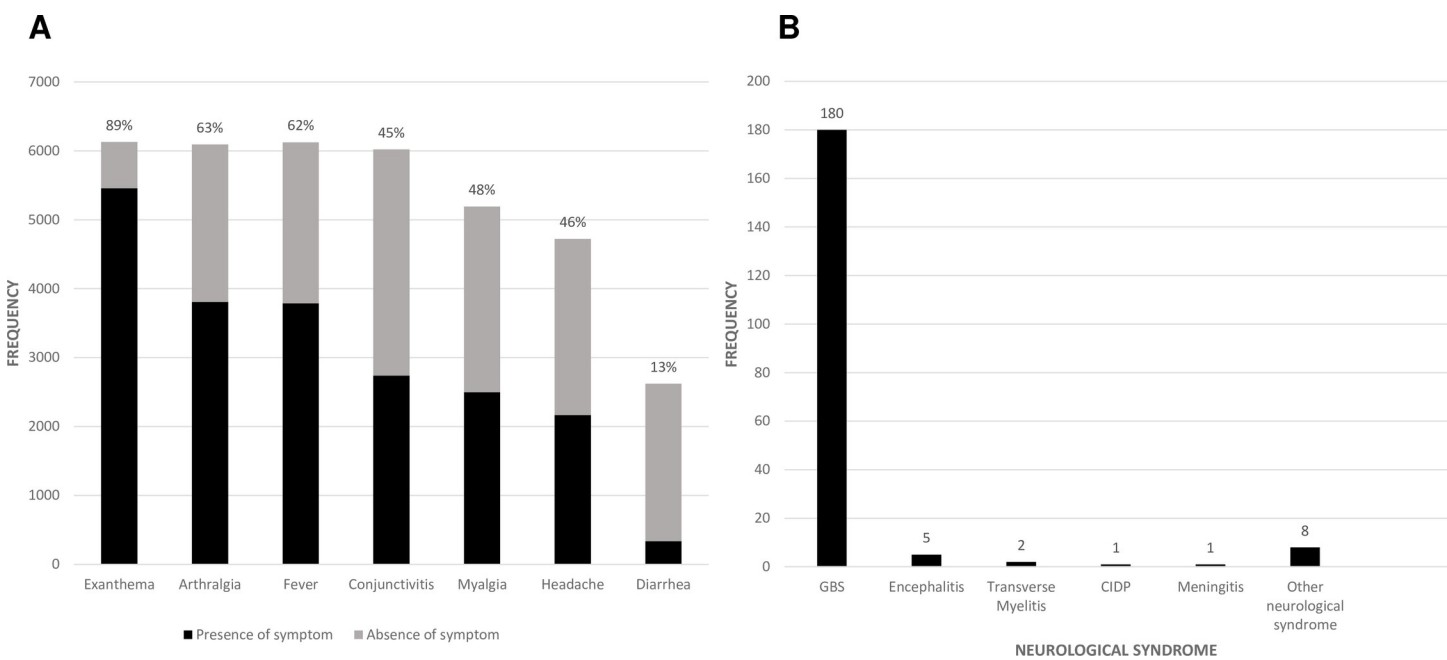

**Fig 2. ZIKV Symptomatology and Neurological Syndromes.** (A) Symptomatology of adults and (B) Neurological syndromes in adults with confirmed ZIKV (similar to WHO criteria) and meeting at least 70% of critical appraisal criteria. CIDP: Chronic inflammatory demyelinating polyneuropathy, GBS: Guillain-Barré Syndrome.

S5 Text includes clinical and laboratory criteria for confirmed ZIKV cases by the authors of the primary articles (relevant to Table 2 of manuscript), and the case definitions of ZIKV by authors of primary articles included in Table 3 of the manuscript.

## ZIKV health outcomes

There were 17,764 subjects identified with confirmed ZIKV infection with confirmatory testing broadly consistent with WHO criteria.

## ZIKV symptomatology, complications, Hospitalizations, and mortality

Symptoms from most common to least common included: exanthema (5,456/6,129; 89%), arthralgia (3,809/6,093; 63%), fever (3,787/6,124; 62%), conjunctivitis (2,738/6,021; 45%), myalgia (2,498/5,192; 48%), headache (2,165/4,722; 46%), and diarrhea (337/2,622; 13%). (Table 2 and Fig 2A) Other reported symptoms and signs from all the studies (beyond those reported in Table 2) included nausea and/or vomiting, lymphadenopathy, pharyngitis, swelling, eye pain, hepatosplenomegaly, and anorexia [52,113].

Many studies had missing data on symptoms, ranging from minor to considerable. This was often related to data collection from patient records in studies with surveillance designs.

Among 27 studies that reported on neurological complications in subjects with laboratory-confirmed ZIKV infection, there were 197 subjects identified with neurological sequelae among 14,496 subjects (1.4%), of which 180 cases were GBS (91%). (Brasil et al. (2016) and Webster-Kerr et al. (2017) were excluded from the analysis as it was unclear if ZIKV was confirmed in the neurological cases [69,92].) Other neurological complications among this cohort included transverse myelitis (2 cases), encephalitis (5 cases), chronic inflammatory demyelinating polyneuropathy (1 case), meningitis (1 case), and other neurological syndromes (8 cases). (Fig 2B)

We considered that 1.4% may be an overestimate for the neurological complication rate given the presence of case-control studies or case series that focused on neurological sequelae. After exclusion of these studies [32,36,42,45,47,53,109], 20 studies remained. Among these, neurological complications occurred in 36 of 14,335 confirmed ZIKV cases (0.3%), of which 27 cases were GBS (75%).

Other neurological manifestations (beyond those reported in Table 2) from the studies include convulsions and optic neuropathy and in particular studies, were reported as distinct neurologic sequelae [82,89].

In terms of hospitalizations, 347 of 3,167 subjects with ZIKV were admitted (11%). This included studies that only recruited hospitalized patients. There were 14 deaths among the 14,202 subjects with ZIKV for whom death was one of the reported outcomes (0.1%). The causes of four of the deaths that we report here were shock and coagulopathy in a patient with vascular comorbidity, hospital-acquired pneumonia, cerebral edema and brainstem herniation in a patient with encephalitis, and septic shock [53,91,109]. In the study by Mercado-Reyes et al., the fatal cases reported are those with co-infection: histopathology on one subject with CHIKV-ZIKV showed tubule-interstitial nephritis, and changes related to systemic inflammatory response syndrome (SIRS), a second case endured multi-organ failure, and a third the histopathology demonstrated acute demyelinating polyneuropathy, pneumonia, and SIRS findings in the liver and spleen.[97] The cause of death for the remaining seven subjects were not reported in the primary studies.

## Guillain-barré syndrome disease incidence and risk factors

Several studies using population-level data reported increases in GBS incidence during the ZIKV epidemic, suggesting a role for ZIKV in GBS pathophysiology. According to Anaya et al. (2017), the incidence of GBS increased 4.41-fold in Cúcuta, Colombia compared to the pre-ZIKV outbreak period [105]. In the Dirlikov et al. study in Puerto Rico, the incidence of GBS in 2016 was 3.5 subjects per 100,000 population, which is 2.1 times greater than the approximate yearly incidence of 1.7 subjects per 100,000[43].

Case-control studies have attempted to characterize GBS in the context of ZIKV infection. Anaya et al. (2017) compared ZIKV-positive subjects with GBS (cases) to ZIKV-positive subjects without GBS (controls) in Cúcuta, Colombia, and reported that lower socioeconomic class or an increased number of previous infections (such as *Mycoplasma pneumoniae*) were two factors associated with increased risk of developing GBS [105]. Dirlikov et al. in 2018 compared 71 subjects diagnosed with GBS with ZIKV and 36 subjects with GBS but without ZIKV in Puerto Rico, and illustrated that subjects with ZIKV more commonly described symptoms of arthralgia and rash than those without ZIKV (arthralgia: 13/71 vs. 1/36 p = 0.03; rash: 36/71 vs. 3/36; p < 0.001) [43]. However, the median duration of seven days from preceding illness to the onset of neurological disease did not differ between the two groups [114]. The Miller Fisher Syndrome, a variant of GBS comprised of a triad of symptoms (ataxia, areflexia, ophthalmoplegia), was present in one subject in each of three studies (Table 4) [42,45,53].

Other studies did not detect an association between ZIKV and GBS. Geurts van Kessel et al. compared GBS cases with healthy controls in Bangladesh and found that the presence of neutralizing antibodies against ZIKV was not significantly increased in GBS cases (odds ratio of 2.23, P = 0.14) [33].

## Guillain-barré syndrome disease outcomes

We report separately the GBS disease progression and outcomes from the subjects across eight studies with confirmed ZIKV (Table 4) and the GBS disease trajectory from subjects across four studies with probable or suspected ZIKV (Table 5).

**Table 4. Outcomes of Guillain-Barré Syndrome Cases that Correspond to ZIKV-Infected Cases with Laboratory-Confirmation of ZIKV Similar to World Health Organization ZIKV-Confirmed Definitions (Corresponds to Table 2).**

| | Number of Subjects with GBS (n) | Time from onset of previous illness to onset of neurologic symptoms, median days | Time from onset of neurological symptoms to nadir, median days | EMG subtype of GBS (AIDP), n of denominator sampled | EMG subtype of GBS (AMAN), n of denominator sampled | Duration of Hospital Stay, median days | Admitted to Intensive Care Unit, n of denominator sampled | Death, n of denominator sampled | Respiratory Failure, n of denominator sampled | Disability or Physical Function Metric | Disability or Physical Function Scores, n |
|---|---|---|---|---|---|---|---|---|---|---|---|
| Cao-Lormeau, 2016, case-control [32] | 41 | 6 | 6 | .. | 41 of 41 | 11 (51 if in intensive care unit) | 16 of 41 | 0 of 41 | 12 of 41 (respiratory assistance) | Ambulation without assistance 3 months post-discharge | 24 of 41 |
| Geurts vanKessel, 2018, case-control [33] | 18 | .. | .. | 5 of 18 | 4 of 18 | .. | .. | 2 of 18 | .. | GBS disability score[115] | 14 of 18 had nadir disability score of 4/5 however 13 could walk independently at 3 months |
| Chang, 2018, case series [42] | 19 | 7 | 5 | 7 of 16 | 2 of 16 | 20 (6 in ICU) | .. | 0 of 19 | .. | Hughes disability score at 1 year | 60% of patients were healthy, 40% with some disability |
| Lynch, 2019, case series [45] | 17 | 10 | .. | .. | .. | 11 (9 in the ICU) | 7 of 8 | 1 of 8 | 3 of 8 (mechanical ventilation) | Recovery | Total recovery: 2 of 8; chronic morbidity: 5 of 8 |
| Uncini, 2018, case series [47] | 20 | 5 | .. | 14 of 20 | 0 of 20 | 31 | 16 of 20 | .. | 12 of 20 respiratory failure (10 had invasive mechanical ventilation, 2 noninvasive mechanical ventilation) | GBS disability scale | At hospital leave, 65% were bedridden or chair bound (grades 4 and 5) |
| da Silva, 2017, cohort study [53] | 27 | 10 | .. | 18 of 27 | 2 of 27 | 8 (0 in the ICU) | 4 of 27 | 1 of 27 | 2 of 27 (mechanical ventilation) | Modified Rankin Scale Score[116] and Hughes GBS Disability Scale score [115] | 3-months: MRS median score 2 (range 1–6) changed by 7 points from nadir. Hughes median 1 (range 0–4). Nineteen of 27 were ambulatory (70%) with 17 (63%) ambulating without assistance. |

*(Continued)*

**Table 4.** (Continued)

| | Number of Subjects with GBS (n) | Time from onset of previous illness to onset of neurologic symptoms, median days | Time from onset of neurological symptoms to nadir, median days | EMG subtype of GBS (AIDP), n of denominator sampled | EMG subtype of GBS (AMAN), n of denominator sampled | Duration of Hospital Stay, median days | Admitted to Intensive Care Unit, n of denominator sampled | Death, n of denominator sampled | Respiratory Failure, n of denominator sampled | Disability or Physical Function Metric | Disability or Physical Function Scores, n |
|---|---|---|---|---|---|---|---|---|---|---|---|
| Hamer, 2017, cross-sectional study [76] | 2 | .. | .. | .. | .. | .. | .. | 0 of 2 | .. | Degree of recovery | One subject had near full recovery, second subject incomplete recovery |
| Gongora-Rivera, 2020, case-control [36] | 11 | -- | -- | 1 of 10 | 1 of 10 | -- | -- | -- | 1 of 10 (mechanical ventilation) | Hughes' functional scale | 3.2 (mean) at nadir |

AIDP = Acute Inflammatory Demyelinating Polyneuropathy, AMAN = Acute motor axonal neuropathy, EMG = Electromyography

**Table 5. Outcomes of Guillain-Barré Syndrome Cases (Corresponds to ZIKV-Infected Cases Depicted in Table 3).**

| | Number of Subjects with GBS (n) | Time from onset of previous illness to onset of neurologic symptoms, median days | Time from onset of neurological symptoms to nadir, median days | EMG subtype of GBS (AIDP), n of denominator sampled | EMG subtype of GBS (AMAN), n of denominator sampled | Duration of Hospital Stay, median days | Admitted to Intensive Care Unit, n of denominator sampled | Death, n of denominator sampled | Respiratory Failure, n of denominator sampled | Disability or Physical Function Metric | Disability or Physical Function Scores, n |
|---|---|---|---|---|---|---|---|---|---|---|---|
| Anaya, 2017, case-control [31] | 29 | 7 | .. | 16 of 27 | 7 of 27 | 23 | 20 of 29 | 0 of 29 | 14 of 20 | Hughes' functional scale [117,118] | 14 of 27 were Class 4 (Bed or chair-bound) at discharge |
| Arias, 2017, case series [40] | 19 | 10 | .. | 0 of 14 | 10 of 14 | 19 in the intensive care unit | 19 | 0 | 15 (respiratory assistance) | Hughes disability score | 15 of 19 scored 4 or 5 at discharge |
| Dirlikov, 2018, case series [43] | 71 | 7 | 7 | 16 of 19 | 2 of 19 | 12 | 44 | 2 | 22 (mechanical ventilation) | Hughes disability score and modified Rankin Scale score at clinical nadir | 4 and 5 median scores respectively |
| Parra, 2016, cross-sectional study [86] | 68 | 7 | .. | 36 of 46 | 1 of 46 | .. | 40 of 68 | 3 of 68 | 21 of 69 (mechanical ventilation) | Median modified Rankin score at nadir | Median score of 4 (IQR 3–5) |
| Rozé, 2017, cross-sectional [87] | 23 | 5.9 | .. | 20 of 23 | 0 of 23 | 60 | 14 of 23 | 2 of 23 | 10 of 23 (respiratory assistance) | Recovery | 1 of 23 cases had recovery |
| Rivera-Correa, 2019, case-control [37] | 7 | 10 | -- | -- | -- | -- | -- | -- | -- | -- | -- |
| Lannuzel, 2019, case series [51] | 38 | -- | 6 | -- | 32 of 36 | -- | 21 of 40 | 1 | 15 of 40 (mechanical ventilation) | Modified Rankin system | 3 scored 1, 11 scored 2–3, 26 scored 4–5 |

IQR = interquartile range

In subjects with confirmed ZIKV, the median number of days from illness onset to onset of neurological symptoms ranged from five to ten days. On electromyography studies, the acute inflammatory demyelinating polyneuropathy subtype (AIDP) was present in 45 of 91 cases (49%) and the acute motor axonal neuropathy (AMAN) subtype was present in 50 of 132 cases tested (38%). Among the probable and suspected cases of ZIKV with GBS for whom electrophysiologic data was available, 88 of 129 (68%) tested subjects presented with the AIDP subtype, while 50 of 132 (38%) tested subjects exhibited the AMAN subtype. (Table 5)

Among this group of confirmed ZIKV and GBS cases, the median length of hospital stay ranged from eight to 31 days. Forty three of 96 cases (45%) were admitted to the intensive care unit (ICU) and the mortality rate among 115 cases of GBS and confirmed ZIKV was 3% (four deaths). The causes of two of these deaths in cases of GBS were septic shock and hospital-acquired pneumonia [53,109]. Respiratory failure and/or mechanical ventilation was reported in 30 of 106 cases of GBS and ZIKV (28%).

Disability and physical function were assessed among studies that reported on GBS using various scoring tools including the GBS disability score, Hughes' functional scale, or modified Rankin scale score [115–118]. In Chang et al. in northern Colombia, 60% (nine of 15) of their subjects with GBS and ZIKV had completely recovered at the one-year mark, and 40% had remaining disability[42]. In contrast, Lynch et al. (2019) in their study in Colombia noted two of eight subjects with GBS and ZIKV had full recovery whereas five of eight subjects had persistent sequelae including weakness, tremors of the face, and sensory deficits [45]. In the Anaya et al., study in Cúcuta, Colombia, dysautonomia predicted poor outcomes, such as resulting disability, in ZIKV-positive GBS cases [105].

Dirlikov et al. in 2018 also noted that subjects with ZIKV and GBS more commonly had difficulty swallowing, paresthesias and weakness of the face, and shortness of breath compared to those with GBS without ZIKV [114]. Also more commonly patients with ZIKV and GBS were admitted to ICU and required ventilatory support than those without ZIKV (ICU admission: 47/71 vs. 16/36 p = 0.03; mechanical ventilation 22/71 vs. 4/36 p = 0.02) [114]. Dirlikov et al. also reported that at six months, it was more common that GBS patients with ZIKV had persistent facial disability compared to those without ZIKV infection [114]. Dirlikov et al. also reported on operative procedures that were required as a result of ZIKV infection included tracheostomy and gastrotomy tube placement [43].

## Further findings

The subsequent sections highlight travel-associated cases, co-infections with ZIKV, the implications of pre-existing health conditions, and pertinent laboratory manifestations of ZIKV.

## Travel-associated cases

Armstrong et al. (2016) reported that among 115 residents of the United States of America with laboratory-confirmed ZIKV infection, 37% had traveled to Central America, 33% to the Caribbean, and 21% to South America with only 6% to Southeast Asia and the Pacific Islands and 2% within North America [66]. Meltzer et al. (2019) described a cohort of Israeli travelers and among the 30 ZIKV-positive cases, 23 of 30 (77%) had traveled to the Americas and 7 of 30 (23%) had traveled to Asia [57]. Of note, in the latter study, there were 248 symptomatic travelers from a total of 1,188 returning Israeli travelers that were tested for ZIKV and only 28 of these 248 (11%) symptomatic travelers were ZIKV-positive [57]. Other travel-related studies demonstrated similar percentages to that summarized here in terms of destinations [78,119].

## Co-Infections

Geurts van Kessel et al. describe cases of GBS and ZIKV-positivity with *Campylobacter jejuni* co-infection (9 of 18 subjects) in Bangladesh. [33] All of the subjects with *C. jejuni* co-infection had an isolated motor presentation of GBS as opposed to the ZIKV-associated GBS cases (without evidence of *C. jejuni* co-infection) in which 6 of 9 cases had a sensory-motor presentation of GBS [109].

Azeredo et al. in 2018 recruited patients in Brazil with suspected arboviral infection in the acute stage with fever, rash, and two other symptoms from a predefined list as well as suspected Zika and dengue cases [120]. Dengue virus (DENV)/ZIKV coinfection occurred in 18 of 134 subjects (13.4%), whereby testing was done using reverse-transcriptase polymerase chain reaction (RT-PCR); DENV mono-infection occurred in 38% and ZIKV mono-infection in 13·4% of cases [120]. Seven subjects had Chikungunya virus (CHIKV) IgM indicating recent infection [120]. To compare symptomatology, the cases that were ZIKV positive consistently reported exanthema and pruritis whereas the DENV-positive subjects often had anorexia, dizziness, vomiting, and prostration [120].

## Comorbidities and pre-existing conditions

Schirmer et al. reported that co-morbidities, including connective tissue disease, dementia, and congestive heart failure in United States Veterans with ZIKV infection were associated with an increased risk of hospitalization [89]. In a cohort study of 101 subjects with human immunodeficiency virus (HIV) infection in Brazil, Calvet et al. measured the CD4+ count and HIV viral loads before ZIKV infection and two months after ZIKV infection and no significant differences were observed [52].

## Laboratory abnormalities in ZIKV

The synthesis of laboratory abnormalities was limited by the variability in reporting. Thrombocytopenia in patients with ZIKV was described in detail by Van Dyne and colleague [48]. Their study in Puerto Rico consisted of 47 subjects with ZIKV-associated thrombocytopenia without another etiology among 37,878 subjects with ZIKV infection (0.1%) [48]. Twelve of these subjects had severe thrombocytopenia (platelet count less than $20 \times 10^9$/L or platelet count less than $50 \times 10^9$/L and clinical management in keeping with a diagnosis of immune thrombocytopenic purpura) and 35 had non-severe thrombocytopenia (platelet count less than $100 \times 10^9$/L that did not meet criteria for severe thrombocytopenia) [48]. Of the subjects with severe thrombocytopenia, all were hospitalized, 33% were admitted to an ICU setting, and mortality was 8% [48].

In the study by Azeredo et al. in Brazil comparing various arboviruses, ZIKV and DENV mono-infections presented with overall lower leukocyte counts compared to cases with no arboviral infections; however, only ZIKV mono-infected subjects show statistically significantly decreased lymphocyte counts compared to non-infected cases [120].

## Discussion

We identified 73 studies globally that reported clinical outcomes in ZIKV-infected adults. Forty of the studies were from the Americas, consistent with the predominance of ZIKV in these countries during the recent epidemic. Travel-associated studies also demonstrated a similar trend in terms of destinations. Of the studies with subjects with confirmed ZIKV and that met at least 70% of critical appraisal criteria, exanthema (5,456/6,129; 89%) and arthralgia (3,809/6,093; 63%) were two common presenting symptoms and 0.3% of infected cases

developed neurologic sequelae, of which 83% were GBS. Several subjects reported recovery from peak of GBS or neurological symptoms; however, some endured chronic disability. Mortality was uncommon, and certain co-morbidities such as heart failure and dementia, as well as complications including GBS and thrombocytopenia, were associated with a greater risk of hospitalization [89].

The frequency of clinical signs and symptoms found by this review, and in particular the high proportions of subjects with fever and rash, is influenced by the clinical case definitions used in the primary studies. Many clinical case definitions of suspected ZIKV infection–especially those developed early in the epidemic–were based on the presence of fever and/or rash with or without additional symptoms. Subsequently, studies have demonstrated that ZIKV infection can occur in the absence of such "cardinal symptoms" and can be minimally symptomatic, and in fact asymptomatic ZIKV infection has long been recognized. Thus, there is bias in the known spectrum of ZIKV clinical features. It should be noted that many studies had considerable missing clinical data due to their retrospective data collection methods, which could have affected estimates. Also important to consider is the integrity of clinical data, which is subject to inaccuracies related to self-reporting and variation in measurement and definitions (e.g., fever).

In this review, there was epidemiologic data supporting an association between adult ZIKV infections and neurological complications, namely GBS, given the mirroring of the trends of these two diseases [32,35,43,49,70,121]. After excluding studies that intentionally enriched for patients with neurological complications, we calculated a risk of neurological sequelae in ZIKV infection of 0.3%. This number remains subject to bias: some of the studies were case series rather than population-based studies, and our requirement for laboratory confirmation of ZIKV infection may have inflated this number as there may be more aggressive testing of severe ZIKV disease. Other systematic reviews and meta-analyses have generated variable estimates. A systematic review and meta-analysis including studies from nine countries until November 2017 showed that 1.23% of ZIKV infections could progress to GBS. [122] Capasso et al. conducted a systematic review and meta-analysis of the GBS incidence rates before and during the ZIKV epidemic and demonstrated that GBS increased 2.6 times during ZIKV over background rates [123]. A meta-analysis of thirty-four studies showed that ZIKV prevalence in GBS was 2.4 to 25 times greater than anticipated, although trends in GBS cases did not mirror fluctuations in ZIKV diagnoses during outbreaks [124]. Specifically regarding subtypes of GBS, the acute inflammatory demyelinating polyneuropathy (AIDP) subtype, classically thought to be related to slowed or decreased conduction speed, was more common among ZIKV-infected subjects than the acute motor axonal neuropathy (AMAN) subtype related to the disintegration of neuronal axons [125]. This has been illustrated by a review by Uncini et al. [126], and is consistent with a meta-analysis of GBS and ZIKV in which the frequency of the AIDP electrophysiologic subtype was 62% followed by 16% [127]. Our data from the probable and suspected ZIKV cases supports this point of AIDP being the more common subtype than AMAN, although ZIKV was not confirmed in these subjects.

Mortality rate in this review was 0.4%. We compare this with a mean case fatality rate of 0.02% from ZIKV illustrated in a systematic review of ZIKV in the Americas. [128] In subjects with confirmed ZIKV and GBS, admission rate to the ICU was 50% and mortality rate was 3%. Consistent with our findings, Leonhard et al. report in all cases, 49% admission rate to ICU and a mortality rate of 1%. [127]

There was significant variability in usage and type of confirmatory testing for ZIKV infection. Given the potential non-specificity of symptoms and overlap with other flaviviruses, other infections, and non-infectious etiologies, subjects with unconfirmed ZIKV infection were not included in our higher-level analysis. Thus, we may have missed true cases of ZIKV

infection and potentially biased the spectrum of disease. We are cognizant that access to confirmatory testing and standardization of testing and case definitions is related to a number of factors, including geographical and site-specific resources and/or where samples could be tested. Some studies compared the results of applying different classification systems to the data to articulate this point [92].

Our systematic review has several limitations. First, the heterogeneity of results was one of the barriers to meta-analyses. This heterogeneity is likely related to multiple factors including the aforementioned variation in clinical case definitions and confirmatory testing.

Second, testing for co-infections and reporting of laboratory abnormalities was heterogenous across studies. The symptoms of ZIKV may mimic those of other arboviruses, which underscores the importance of delineating a mono-infection from co-infection with another arbovirus and from cross-reactivity in serologic testing. In several studies, enzyme-linked immunosorbent assay (ELISA) IgM results for other arboviruses were IgM or IgG positive [33,41,44]. However, the interpretation of these results requires other more definitive methodologies. Schirmer et al. used an appropriate testing algorithm, in which specimens where ZIKV RT-PCR was negative or not done, specimens were further tested with IgM ELISAs for ZIKV and if positive, equivocal or inconclusive, they were subjected to plaque reduction neutralization testing (PRNT) for the suspected virus or viruses [89]. We did not summarize data on co-infections versus cross-reactivity given the challenges with consistent testing algorithms. Biochemical abnormalities are potentially overestimated or underestimated as well. For instance, in the study by Van Dyne et al. that described ZIKV-associated thrombocytopenia, 28% of the charts of patients that reported thrombocytopenia were available for review [48]. In future studies, consistent co-infection testing and surveillance for laboratory abnormalities will be required to accurately estimate the incidences of these outcomes.

Third, categorization of studies into study type (cross-sectional, cohort, case series, case-control) was challenging. Some of the studies were re-categorized by the systematic review authors compared to how the studies self-described. For example, we re-labelled studies that were described as prevalence as cross-sectional studies if they reported on both exposures and outcomes but did not have features of cohort studies, case-control studies, or case series [71]. This was in keeping with the description of analytical cross-sectional studies as described by Alexander et al [129]. One of the included studies used a mixed-methods approach, thus for classification purposes we chose the dominant study type to report in our systematic review [105].

Fourth, the description of GBS disease progression and outcomes, including admission to ICU and mechanical ventilation, are highly dependent on geographic location and available hospital resources [130]. Moreover, comorbidities play a role in hospitalization and death and can positively or negatively influence the likelihood of receiving ZIKV diagnosis and we recognize the complex interplay of these factors on the total numbers [56,89,91]. We recognize also that the percentage of hospitalization in ZIKV patients reported in our study (16%) may be inflated by the inclusion of case series of hospitalized patients.

Finally, some studies had mixed populations including adults, children, and congenital cases. We defined the adult population as 18 years of age and older, but some studies defined the adult age group differently, and pediatric cases could not be separated for reporting. Children and congenital cases were removed from final counts when possible, or indicated where this was not possible [57,69,75–77,80,84,88,105,107].

The strengths of our systematic review are the thorough and comprehensive search strategy employed, including studies until September 2020, the use of broad keywords of "Zika virus" and "Zika infection," and the lack of language restrictions which allowed us to include as many studies as possible. After exclusion criteria and removing children and CZS-based

studies, there was a large body of literature to extract data from. Several studies have reported on the virology, testing, and differential diagnoses for ZIKV; however, to our knowledge this is the first systematic review to synthesize the epidemiology, symptomatology and outcomes of adult ZIKV infection globally [131–133]. This study contributes both to the body of clinical epidemiology literature of ZIKV and to that of travel medicine. Our results have depicted the geographic distribution of cases as well as those that are travel-related, and highlighted risk factors for developing complications and hospitalization associated with ZIKV infection.

## Supporting information

**S1 PRISMA Checklist.**
(DOC)

**S1 Table. Study Type Classifications.** Included in this supplement is a table categorizing each study by study type ('case-control,' 'case series,' 'cross-sectional,' or 'cohort') and further sub-categorizing each study into 'surveillance,' 'public health-based,' or 'other' (hospital-based, single-center). CDC = Centre for Disease Control, GBS = Guillain-Barré Syndrome, ICU = intensive care unit, INS = Instituto Nacional de Salud (in Colombia), US = United States WHO = World Health Organization
(DOCX)

**S1 Fig. Classification of Included Studies in the Systematic Review.** (A) Classified by study-type (classified by authors of systematic review), (B) Classified by geographic location of subjects and (C) Classified by involvement in public health reporting or surveillance versus other (purely hospital-based, healthcare center-based, population-based, travel-clinic based).
(TIF)

**S2 Fig. Diagrammatic Representation of Distribution of Primary Studies.** This figure depicts the grouping of primary studies into tables within our manuscript based on ZIKV testing methodologies and details included within each primary study.
(TIF)

**S1 Text. Full Search Strategy for Systematic Review.** Included here the search strategy and review process with details on the searches performed from five of the included databases.
(DOCX)

**S2 Text. PROSPERO protocol CRD 42018096558 used for this study.** Protocol amendment has been submitted to include three authors on the protocol.
(PDF)

**S3 Text. Joanna Briggs Institute (JBI) Critical Appraisal Tool and ZIKV Adult Population Results.** S3A. Table JBI Critical Appraisal Tool Questionnaire for Case-Control Studies Applied to ZIKV Systematic Review. S3B Table. JBI Critical Appraisal Tool Questionnaire for Case Series Applied to ZIKV Systematic Review. S3C Table. JBI Critical Appraisal Tool Questionnaire for Cohort Studies Applied to ZIKV Systematic Review. S3D Table. JBI Critical Appraisal Tool Questionnaire for Cross-Sectional Studies Applied to ZIKV Systematic Review. Note: Causation cannot be inferred from cross-sectional studies, though if no statistical analysis was performed, a point was subtracted from the critical appraisal for the particular study.
(DOCX)

**S4 Text. Summary of Data Processing for Adult ZIKV Clinical Manifestations and Health Outcomes.**
(DOCX)

**S5 Text. ZIKV Case Definitions by Authors of Primary Articles. S5A Table–Clinical and laboratory criteria for Confirmed ZIKV Case by Authors from Primary Articles Included in Table 2 in Manuscript.** DENV = Dengue Virus, PAHO = Pan American Health Organization, PRNT = Plaque Reduction Neutralization Test, RT-PCR = Reverse Transcriptase-Polymerase Chain Reaction, RNA = Ribonucleic Acid, VNT = Virus Neutralization Test, WHO = World Health Organization. **S5B Table–Case Definitions of ZIKV by Authors of Primary Articles Included in Table 3 in Manuscript.** DENV = Dengue Virus, ECDC = European Center for Disease Control Clinical Case Definition = ZIKV infection defined as maculopapular rash with or without fever, and painful joints or muscles or non-purulent conjunctivitis, GBS = Guillain Barré Syndrome, ELISA = Enzyme-Linked Immunosorbent Assay, INS = National Health Institute, PAHO = Pan American Health Organization [45], PCR = polymerase chain reaction, RNA = Ribonucleic Acid, RT-PCR = Reverse Transcriptase-Polymerase Chain Reaction, rRT-PCR = Real-time Reverse Transcriptase-Polymerase Chain Reaction, VNT = Virus Neutralization Test, ZIKV = Zika Virus, ZVD = Zika Virus Disease.
(DOCX)

## Author Contributions

**Conceptualization:** Shaun K. Morris, Kellie E. Murphy, Beate Sander.

**Data curation:** Joanna M. Bielecki.

**Formal analysis:** Sheliza Halani, Panashe E. Tombindo, Ryan O'Reilly, Laura K. Erdman, Clare Whitehead, Lauren Ramsay, Justin Boyle, Carsten Krueger, Shannon Willmott.

**Funding acquisition:** Beate Sander.

**Investigation:** Sheliza Halani, Panashe E. Tombindo, Ryan O'Reilly, Rafael N. Miranda, Laura K. Erdman, Clare Whitehead, Joanna M. Bielecki, Lauren Ramsay, Raphael Ximenes, Justin Boyle, Carsten Krueger, Shannon Willmott.

**Methodology:** Sheliza Halani, Panashe E. Tombindo, Ryan O'Reilly, Rafael N. Miranda, Laura K. Erdman, Clare Whitehead, Joanna M. Bielecki, Lauren Ramsay, Raphael Ximenes, Justin Boyle, Carsten Krueger, Shannon Willmott, Shaun K. Morris, Kellie E. Murphy, Beate Sander.

**Project administration:** Rafael N. Miranda.

**Resources:** Beate Sander.

**Software:** Rafael N. Miranda.

**Supervision:** Shaun K. Morris, Kellie E. Murphy, Beate Sander.

**Validation:** Shaun K. Morris, Kellie E. Murphy, Beate Sander.

**Visualization:** Sheliza Halani, Panashe E. Tombindo, Rafael N. Miranda, Laura K. Erdman, Raphael Ximenes, Beate Sander.

**Writing – original draft:** Sheliza Halani, Panashe E. Tombindo, Beate Sander.

**Writing – review & editing:** Sheliza Halani, Panashe E. Tombindo, Ryan O'Reilly, Rafael N. Miranda, Laura K. Erdman, Clare Whitehead, Joanna M. Bielecki, Lauren Ramsay, Raphael Ximenes, Justin Boyle, Carsten Krueger, Shannon Willmott, Shaun K. Morris, Kellie E. Murphy, Beate Sander.

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
