## [Decision Letter · Decision Letter 0]

12 Sep 2020

Dear Dr. Halani,

Thank you very much for submitting your manuscript "Clinical Manifestations and Health Outcomes Associated with Zika Virus Infections in Adults: A Systematic Review" for consideration at PLOS Neglected Tropical Diseases. As with all papers reviewed by the journal, your manuscript was reviewed by members of the editorial board and by several independent reviewers. In light of the reviews (below this email), we would like to invite the resubmission of a significantly-revised version that takes into account the reviewers' comments. 

We cannot make any decision about publication until we have seen the revised manuscript and your response to the reviewers' comments. Your revised manuscript is also likely to be sent to reviewers for further evaluation.

Sincerely,

Pedro F. C. Vasconcelos

Deputy Editor

Pedro Vasconcelos

Deputy Editor

Reviewer's Responses to Questions

**Key Review Criteria Required for Acceptance?**

**Methods**

-Are the objectives of the study clearly articulated with a clear testable hypothesis stated?

-Is the study design appropriate to address the stated objectives?

-Is the population clearly described and appropriate for the hypothesis being tested?

-Is the sample size sufficient to ensure adequate power to address the hypothesis being tested?

-Were correct statistical analysis used to support conclusions?

-Are there concerns about ethical or regulatory requirements being met?

Reviewer #1: A meta-analysis would be of paramount importance in this manuscript, but authors addressed it was not possible due primary studies limitations. Please see my comments in the attached file.

Reviewer #2: In general, the study designs are appropriate. The only issue is the timing of when the searches stopped. The study objectives are clear and the means taken to eliminate some extraneous data are well-described.

Reviewer #3: The study is aimed at describing the clinical manifestations of ZIKV infection in adults performing a systematic review of observational studies and clinical trials. One of the main challenges described by the authors regards the heterogeneity of such studies, with focus being reported on the diagnostic classification of patients. However, with such a high variation of studies I missed a more comprehensive approach and discussion on the clinical signs and symptoms evaluation as these also can suffer from subjective and diverse methodologies applied. In this aspect I believe the manuscript would benefit of a more comprehensive description on the completeness of information regarding the symptoms presented, intensity and timeline of symptoms emergence.

One important aspect that was only briefly mentioned by the authors regards the inclusion/eligibility criteria. It should be mentioned that studies using fever and or any symptom as a condition for inclusion will have its estimates biased. It is important to be more descriptive of such symptoms as many guidelines try to differentiate arbovirus infections manifestations based on the prevalence and intensity of such manifestations. ZIKV has been described as causing a high proportion of asymptomatic infections, based largely on retrospective survey data that could be disputed. 

Regarding the clinical outcomes, mainly regarding neurological manifestations, it is important to mention the study design specifically in the text and tables as this is also influential on how the data is interpreted. The assertion present in the abstract that 6% of cases developed neurologic sequelae is misleading of the real incidence of such complication and must be reviewed.

**Results**

-Does the analysis presented match the analysis plan?

-Are the results clearly and completely presented?

-Are the figures (Tables, Images) of sufficient quality for clarity?

Reviewer #1: I pointed some improvements that are necessary to clarify the results. Please see my comments in the attached file.

Reviewer #2: Their review of the literature is adequate and appropriate.

Reviewer #3: There are issues regarding the presentation of the results that are related to the data collected and the limitations of the heterogeneous reporting. Although the authors refrained of performing a meta-analysis, the abstract presents broad frequencies that can be misleading. The review of clinical manifestations has the possibility of providing a broad and comprehensive overview of the clinical picture of this infection, however the detail on the manifestations and the limitations of how the data is reported hamper this contribution that could lay ground for building the need for standardizing the clinical studies data collection and reporting for such studies

**Conclusions**

-Are the conclusions supported by the data presented?

-Are the limitations of analysis clearly described?

-Do the authors discuss how these data can be helpful to advance our understanding of the topic under study?

-Is public health relevance addressed?

Reviewer #1: Yes, authors discussed conclusions and limitations properly.

Reviewer #2: This is a review to their conclusions aren't based upon their own data. However, the discussion does a good job putting their review into context and offers thoughtful caveats and limitations for their interpretations.

Reviewer #3: The authors report some of the main limitations but, in my opinion, fail to highlight the high variability of inclusion criteria, clinical signs and symptoms ascertainment and outcomes measurement.

It is very positive that they shed light on the need to investigate co-infections and describing other needed areas for standardization of the reports.

**Editorial and Data Presentation Modifications?**

Reviewer #1: I addressed some points in my attached review.

Reviewer #2: Overall the data are well summarized. A number of clarifications can be made to enhance the manuscript but are minor. There is one major concern which is addressed elsewhere.

Reviewer #3: (No Response)

**Summary and General Comments**

Reviewer #1: Please see my comments in the attached file.

Reviewer #2: Major Concerns

This article is being reviewed in August/September of 2020, but the most recent update to data was performed in December 2018. What is the reason for why a more recent search has not been made to update these findings? It is very likely that most relevant information was reported prior to 2018, but some reports of less likely Zika manifestations might have been reported after 2018. Further, it would help differentiate this review from others to have a more recent dataset.

Minor Concerns

Lines 66-67. Vector-borne infections are from viruses, but “dengue” and “chikungunya” refer here to the disease. Please modify to state “vector-borne diseases.”

Line 62, 71. ZIKV was used before it was defined in line 71. Please define in line 62.

It might read better if paragraphs 2 and 3 in the Introduction were combined.

Lines 161, 162 and several places thereafter. Why are five and eight written out while 11 and 29 numbers? Usually the number name is only written if it begins a sentence. I would recommend listing all in numbers or number names.

Reviewer #3: In this study the authors describe the results of a systematic review performed to describe the signs and symptoms of ZIKV infection. This is an important area and topic as there is a broad variation of reports, including the estimates of asymptomatic infections and complications and that ZIKV emergence in Latin America was overlooked for a long period due to overlap of symptoms with other common etiologies of febrile illness. The methods for selecting studies are clear and well applied but some improvement could be done regarding the clinical aspects and reporting of the studies, which could improve the interpretation and relevance of the data.

PLOS authors have the option to publish the peer review history of their article (what does this mean?). If published, this will include your full peer review and any attached files.

Reviewer #1: No

Reviewer #2: No

Reviewer #3: No
---

## [Decision Letter · Decision Letter 1]

28 May 2021

Dear Dr. Halani,

We are pleased to inform you that your manuscript 'Clinical Manifestations and Health Outcomes Associated with Zika Virus Infections in Adults: A Systematic Review' has been provisionally accepted for publication in PLOS Neglected Tropical Diseases.

Best regards,

Pedro F. C. Vasconcelos

Deputy Editor

Pedro Vasconcelos

Deputy Editor

Reviewer's Responses to Questions

**Key Review Criteria Required for Acceptance?**

**Methods**

-Are the objectives of the study clearly articulated with a clear testable hypothesis stated?

-Is the study design appropriate to address the stated objectives?

-Is the population clearly described and appropriate for the hypothesis being tested?

-Is the sample size sufficient to ensure adequate power to address the hypothesis being tested?

-Were correct statistical analysis used to support conclusions?

-Are there concerns about ethical or regulatory requirements being met?

Reviewer #3: The study has clear objectives laid down and applies comprehensive and extensive methods to provide an abragent and detailed review of Zika infection manifestations. The revised version is much improved and provides a clear and complete description of the results.

**Results**

-Does the analysis presented match the analysis plan?

-Are the results clearly and completely presented?

-Are the figures (Tables, Images) of sufficient quality for clarity?

Reviewer #3: The results are well presented and acoording with the methods applied. Presentation has improved considerably.

**Conclusions**

-Are the conclusions supported by the data presented?

-Are the limitations of analysis clearly described?

-Do the authors discuss how these data can be helpful to advance our understanding of the topic under study?

-Is public health relevance addressed?

Reviewer #3: The conclusions have been improved in the new version of the manuscript and in line with the presented results.

**Editorial and Data Presentation Modifications?**

Reviewer #3: None

**Summary and General Comments**

Reviewer #3: The manuscript presents the data of a comprehensive and well conducted systematic review on the clinical manifestations of Zika infection which is very useful and can be used to inform clinicians and researchers working with this disease.

PLOS authors have the option to publish the peer review history of their article (what does this mean?). If published, this will include your full peer review and any attached files.

Reviewer #3: **Yes: **Andre Siqueira

---

## [Editor Report · Acceptance letter]

30 Jun 2021

Dear Dr. Halani,

We are delighted to inform you that your manuscript, "Clinical Manifestations and Health Outcomes Associated with Zika Virus Infections in Adults: A Systematic Review," has been formally accepted for publication in PLOS Neglected Tropical Diseases.

Best regards,

Shaden Kamhawi

co-Editor-in-Chief

Paul Brindley

co-Editor-in-Chief
